# Those Who Have Continuing Radiation Anxiety Show High Psychological Distress in Cases of High Post-Traumatic Stress: The Fukushima Nuclear Disaster

**DOI:** 10.3390/ijerph182212048

**Published:** 2021-11-17

**Authors:** Masatsugu Orui, Chihiro Nakayama, Nobuaki Moriyama, Masaharu Tsubokura, Kiyotaka Watanabe, Takeo Nakayama, Minoru Sugita, Seiji Yasumura

**Affiliations:** 1Department of Public Health, Fukushima Medical University School of Medicine, Fukushima 960-1295, Japan; nakac@fmu.ac.jp (C.N.); moriyama@fmu.ac.jp (N.M.); yasumura@fmu.ac.jp (S.Y.); 2Sendai City Mental Health and Welfare Center, Sendai 980-0845, Japan; 3Department of Radiation Health Management, Fukushima Medical University School of Medicine, Fukushima 960-1295, Japan; tsubokura_tky@me.com; 4Department of Medicine, Teikyo University School of Medicine, Tokyo 173-8606, Japan; kiyowata@med.teikyo-u.ac.jp; 5Department of Health Informatics, School of Public Health, Kyoto University, Kyoto 606-8501, Japan; nakayama.takeo.4a@kyoto-u.ac.jp; 6Toho University, Tokyo 143-8540, Japan; sugitamnr@a05.itscom.net

**Keywords:** psychological distress, post-traumatic stress, anxiety, media, Fukushima Daiichi nuclear power plant accident

## Abstract

Background: this cross-sectional study aimed to clarify the associations among media utilization, lifestyles, and the strong radiation anxiety that has persisted 9 years after the 2011 nuclear accident. Moreover, the relationships among psychological distress, post-traumatic stress, and strong radiation anxiety were examined. Methods: for the multivariate regression analysis, the independent variables were radiation anxiety at the time of the accident and the current status, categorized as “continuing/emerging strong radiation anxiety”. Media utilization (local, national, internet, and public broadcasts, and public relations information) and lifestyle variables (sleep quality, regular exercise, and drinking habits) were set as the dependent variables. Moreover, the psychological distress of residents with continuing/emerging strong radiation anxiety was examined by an analysis of covariance stratified by post-traumatic stress. Result: there was no significant association between lifestyle variables and media utilization, except for local media (OR: 0.435, 95% CI: 0.21–0.90). Conversely, significantly high psychological distress was confirmed among residents with continuing/emerging radiation anxiety. The K6 score, representing psychological distress, for those with higher post-traumatic stress was 12.63; for those with lower post-traumatic stress, it was 5.13 (*p* = 0.004). Conclusions: residents with continuing/emerging strong radiation anxiety showed high psychological distress, which has been strengthened by higher post-traumatic stress.

## 1. Introduction

The Great East Japan Earthquake (GEJE) occurred on 11 March 2011, triggering a massive tsunami that led to enormous damage along the Pacific coast in Japan. Following this, a separate tsunami hit the Fukushima Daiichi nuclear power plant, resulting in a nuclear disaster in Fukushima Prefecture. According to previous reports from officials, there were no direct cases of death from low-dose radiation exposure [1]. However, ongoing research regarding morbidity related to a particular cancer is required to identify the potential associations between exposure and absorbed doses and the risk of developing cancers [2].

The nuclear accident at the Fukushima Daiichi nuclear power plant caused multiple public health problems, one of which was increased anxiety and mental health issues due to perceived risk among the evacuees and residents of Fukushima. Our previous study reported that the current status of psychological distress, post-traumatic stress, and radiation health anxiety remained high for those rebuilding their permanent homes after evacuation orders had been lifted [3]. However, it is necessary to indicate a clear interaction among high psychological distress, continuing strong radiation anxiety, and post-traumatic stress.

We supposed that media utilization could be one of the factors that could have affected continuing strong radiation anxiety. Yamashita, who supported the nuclear accident response efforts on-site as a radiation specialist, argued that the Fukushima event was not only a health disaster but also an information disaster [4], because the accident was an unprecedented experience for residents and their perceived radiation exposure risk may have been related to the mass media. Consequently, their disaster-related stress and/or psychological distress levels may have been affected [5]. Indeed, newspaper coverage of the accident focused mainly on the crisis response related to immediate issues, actions, and decisions in the aftermath of the accident (e.g., information regarding on-site actions undertaken, communications about the INES (International Nuclear Event Scale), food restrictions, costs, and the number of people affected and being evacuated) [6]. One of the recommendations of the Chernobyl Forum report was to address the lack of accurate information available to local populations on the health risks [7]. Moreover, the United Nations Sendai Framework for Disaster Risk Reduction aimed to help people understand the disaster risk by sharing non-sensitive information and appropriate communications, and to strengthen the utilization of media, including social media and traditional media [8]. In fact, the media functioned as a form of interpersonal communication with others or as channels for local governments, organizations, and local and mass media during the immediate aftermath of the Great East Japan Earthquake [8]. Our previous studies reported that local TV broadcasts or public relations information from local governments was associated with lower health anxiety due to radiation [9,10].

However, the media are not always helpful. Several studies have examined disaster-related television viewing in the context of terrorism and have explored a range of outcomes, including post-traumatic stress disorder (PTSD), depression, anxiety, stress reactions, and substance use [11]. One study reported a significant association between the consumption of television and internet coverage of the 2011 Great East Japan Earthquake and the associated tsunami and post-traumatic reactions [12]. This suggests that media use (including the internet and television) can trigger negative psychological responses in evacuees and residents who use the media.

Following the occurrence of the GEJE and the Fukushima Daiichi nuclear power plant accident, numerous health promotion activities have been carried out to help residents with any health issues, including mental health concerns (e.g., encouraging regular exercise, good quality sleep, and appropriate drinking through disaster-related health activities and public awareness), based on previous review articles or longitudinal studies [13,14,15,16]. Therefore, such health promotion activities might be a clue to resolving whether strong health-related radiation anxiety has any association with reducing the risk of continued strong radiation anxiety.

Against this backdrop, we formed two hypotheses. First, continuing and emerging strong health-related radiation anxiety may be associated with specific media utilization or undesirable lifestyles, such as poor-quality sleep, no regular exercise habits, and inappropriate drinking. Second, those who have continuing radiation anxiety may show high psychological distress, which may also be strengthened by traumatic events. Therefore, the present study aimed to clarify the associations among media utilization, undesirable lifestyles, and strong health-related radiation anxiety 9 years after the accident. Moreover, we examined the relationship between psychological distress and post-traumatic stress among those who have strong health-related radiation anxiety. These findings are likely to offer useful tips for risk reduction among residents who have been experiencing strong long-lasting health-related radiation anxiety.

## 2. Materials and Methods

### 2.1. Study Design and Participants

The cross-sectional questionnaire survey (Appendix A) targeted 1600 residents of the Fukushima Prefecture aged between 20 and 79 years. We selected 400 people from the evacuation area designated by the Japanese government, comprising Tamura City, Minami-Soma City, Kawamata Town, Hirono Town, Naraha Town, Tomioka Town, Okuma Town, Futaba Town, Namie Town, Kawauchi Village, Katsurao Village, and Iitate Village. As a sample from the non-evacuation area, 1200 residents from three other areas (Hama-Dori, Naka-Dori, and Aizu) were selected, 400 people from each area (Figure 1). Participant selection was based on a two-stage stratified random sampling (Stage 1 was a survey of the region; Stage 2 was a survey of individuals). Approximately 30 to 31 individuals per area were randomly selected from municipal resident registration files to obtain 1600 representative participants. We sent an anonymous, self-reporting postal questionnaire to the subjects in January 2020.

### 2.2. Survey Variables

#### 2.2.1. Health-Related Radiation Anxiety at the Time of the Accident and Currently

For anxiety levels regarding radiation health risks, participants were asked to subjectively rate their current level of anxiety about the effects of radiation on their health on a five-point scale ranging across “not at all”, “only a little”, “somewhat”, “very”, and “extremely”. Moreover, participants were asked about their level of anxiety about the effects of radiation on their health at the time of the nuclear power plant accident. Those giving responses of “very” and “extremely” were categorized as the “strong anxiety” group, while other levels of anxiety were categorized as the “no or weak anxiety” group, based on a previous study [12]. This questionnaire was designed by the investigators.

#### 2.2.2. Media Utilization

Based on our previous study, regarding their utilization of media to obtain information about radiation, respondents selected up to three items from the following 13 options: local newspapers, national newspapers, NHK television (i.e., public broadcast television, both national and local), private local broadcast television, private national broadcast television, radio, internet news, internet sites/blogs, social network services (SNSs), magazines/books, public relations information from local governments, word of mouth, and none of the above. These options were categorized as “any local media” (local newspapers and broadcasting), “any national media” (national newspapers and broadcasting), “public broadcasting” (NHK), “any internet media” (internet news, internet sites/blogs, SNSs), and “public relations information from local governments” and used as dependent variables to assess the association between media utilization and current strong health anxiety [11,12].

#### 2.2.3. Lifestyle Variables Related to Health-Promoting Activities Following the Accident

For regular exercise, participants were asked to subjectively rate how many times they carried out physical activities and exercise on average a month on a five-point scale ranging across “not at all”, “one to three times”, “four to seven times”, “8 to 15 times”, and “more than 15 times”. Those giving responses of “8 to 15 times” and “more than 15 times” were categorized as the “regular exercise” group. This definition was based on the National Health Promotion guidelines of the Japanese Ministry of Health, Labor, and Welfare.

For the current sleep quality, participants were asked to subjectively rate how satisfied they were with the quality of their sleep condition in the past month on a four-point scale, choosing between “satisfied”, “a little dissatisfied”, “quite dissatisfied”, and “extremely dissatisfied”.

For drinking behavior, participants were asked to subjectively rate whether they drank a particular amount of alcohol every day: 500 mL of beer, 1 go (180 mL) of Japanese sake, or two glasses of wine, with the possible answers being “yes”, “no”, and “never drink or quit drinking”. Notably, 500 mL of beer, 180 mL of Japanese sake, and 240 mL of wine are equivalent to two drinks, an amount of alcohol that is defined as “appropriate drinking” by the National Health Promotion guidelines of the Japanese Ministry of Health, Labor, and Welfare.

This questionnaire regarding lifestyle was also designed by the investigators. Regular exercise, being aware of sleep quality, and encouraging appropriate drinking behavior were suggested as health-promoting activities following the Fukushima Daiichi nuclear power plant accident and as ordinary health-promoting activities. According to previous studies, these variables were associated with residents’ mental health status in Fukushima. Therefore, we selected these variables as independent variables in the present study [13,17,18].

#### 2.2.4. Current Mental Health Status

The Kessler 6 (K6) scale and the four-item PTSD checklist (four-item PCL) were used to measure the current mental health status to assess psychological distress and post-traumatic stress among residents in Fukushima. The K6 scale is utilized to screen for non-specific serious mental illnesses, including DSM-IV mood and anxiety disorders, indicating psychological distress within the last 30 days. The score on the K6 scale ranged from 0 to 24 points. Those scoring 0–4 points were classified as probably having no psychological distress, and those scoring 5–12 points were classified as having probable mild to moderate stress. Meanwhile, a score of 13+ points was defined as serious psychological distress [19]. This study used the Japanese version of the K6 scale, which has been empirically validated as an independent means of screening for mental distress among evacuees [20,21].

The four-item PCL was originally based on the PTSD Checklist-Specific (PCL-S) comprising 17 items assessing PTSD symptoms, all of which are rated on a Likert scale, ranging from 1 = “not at all” to 5 = “extremely” [22]. The Japanese version of the PCL-S has previously been validated [23,24]. In this study, we used the abbreviated version of the PCL-S, which comprised four measurements (re-experiencing: repeated, disturbing, and unwanted memories of the stressful experience; physical reactions: having strong physical reactions when something reminds the respondent of the stressful experience, such as heart pounding, trouble breathing, and sweating; avoidance: avoiding external reminders of the stressful experience; and difficulty concentrating). The score for the four-item PCL can range from 4 to 20 points, and the checklist’s reliability and validity has been proved [25,26].

### 2.3. Statistical Analysis

A chi-square test was used to examine a simple tabulation of the basic characteristics, health-related radiation anxiety, media utilization for information about radiation, and lifestyle variables related to health-promoting activities following the accident. Moreover, a *t*-test was performed to compare both psychological distress and post-traumatic stress in residents with strong radiation anxiety. Furthermore, a logistic regression analysis was used to examine the association between strong radiation anxiety, media utilization for information radiation, and lifestyle variables related to health-promoting activities while adjusting for age, gender, educational background, and the four different areas of Fukushima Prefecture. Finally, the psychological distress status of those who had continuing strong radiation anxiety was examined by analysis of covariance (ANCOVA), stratified according to higher/lower strong post-traumatic stress levels. The statistical significance was evaluated using two-sided design-based tests with a 5% significance level with Stata 15 (StataCorp, 2017. Stata Statistical Software: Release 15. College Station, TX, USA: StataCorp LLC).

A conceptual model of the analysis in the present study is shown in Figure 2.

### 2.4. Ethical Consideration

The ethical review committee of Fukushima Medical University approved the survey on 16 July 2019 (approval number 2019-110). This questionnaire survey was mailed to randomly selected subjects, and they were asked to answer anonymously. Therefore, informed consent could not be obtained in advance. It was considered that the respondents agreed to participate in the survey if they answered the questionnaire.

## 3. Results

### 3.1. Participants

We sent out 1600 questionnaires to residents of Fukushima from January to March 2020 and received 737 responses (i.e., a response rate of 46.1%). We excluded 42 respondents who failed to provide information regarding their gender, age, and living area because this basic demographic information was essential for the analysis of this survey. The final study population consisted of 695 respondents, whose inclusion was determined by analyzing their basic characteristics. For the analysis of strong health-related radiation anxiety, media utilization, and lifestyle variables related to health-promoting activities and post-traumatic stress, we excluded 17 respondents who did not answer the questions about their health-related radiation anxiety (Figure 3).

### 3.2. Basic Characteristics, Media Utilization, Lifestyle Variables, and Current Mental Health Status of Respondents

The majority of respondents from Fukushima were aged 65 years and above and had a junior or senior high school education background among respondents (Table 1).

Regarding media utilization for information about radiation, local media was the most utilized, and the second most utilized was public broadcasting (NHK). For the lifestyle variables, the proportion of subjects with a regular exercise habit (more than twice a week) was 10.4%, the proportion who were satisfied with their sleep was 32.8%, and those with appropriate drinking habits made up 29.4%. Regarding the current mental health status, the mean K6 score was 4.08 (standard deviation (SD): 0.18), and the mean four-item PCL score was 5.82 (SD: 0.10) (Table 2).

### 3.3. Definition and Characteristics of Continuing, Emerging, Improved, and No Strong Radiation-Related Health Anxiety

We focused on the possible change in health-related radiation risk perceptions from the time of the accident to the current status. We defined “continuing”, “emerging”, “improved”, and “no” strong health-related radiation anxiety as follows:Continuing strong health-related radiation anxiety was defined as an answer of “extremely” or “very” regarding the level of anxiety about effects of radiation on health both at the time of the accident and currently.Emerging strong health-related radiation anxiety was defined as an answer of “somewhat”, “only a little”, or “not at all” regarding the level of anxiety about the effects of radiation on health at the time of the nuclear power plant accident and an answer of “extremely” or “very” regarding current anxiety about the effects of radiation on health.Improved strong health-related radiation anxiety was defined as an answer of “extremely” or “very” regarding the level of anxiety about the effects of radiation on health at the time of the nuclear power plant accident and an answer of “somewhat”, “only a little”, or “not at all” regarding the level of current anxiety about the effects of radiation on health.No strong health-related radiation anxiety was defined as an answer of “somewhat”, “only a little”, or “not at all” regarding the level of anxiety about the effects of radiation on health, both at the time of the accident and currently.

As a result of this categorization, 42 respondents were categorized as having continuing strong health-related radiation anxiety, 6 respondents were categorized as having emerging strong health-related radiation anxiety, 159 respondents were categorized as having improved strong health-related radiation anxiety, and 471 respondents were categorized as having no strong health-related radiation anxiety. Incidentally, in this analysis, continuing and emerging strong health-related radiation anxiety were combined into a single variable because the number of respondents with emerging strong radiation anxiety was small. Moreover, they were potentially the main targets of mental health support in the current situation and following the accident. The proportion of those with continuing/emerging strong radiation anxiety was 7.1% (95% confidence interval, CI: 5.2–9.0) (Table 3).

Regarding the basic characteristics of those with continuing and emerging strong health-related radiation anxiety, women, those with higher educational backgrounds, and those living in the evacuation area and Naka-Dori had higher proportions (Table 4).

### 3.4. Simple Tabulation of Media Utilization, Lifestyle Variables, and Current Mental Health Status

For the media utilization and lifestyle variables, there were no significant differences among health-related radiation anxiety levels. On the other hand, for respondents with continuing/emerging strong health-related radiation anxiety, the mean scores of the K6 and four-item PCL were significantly higher than those in other categories (K6 score: 8.13 (SD: 1.06), four-item PCL: 7.70 (0.60)) (Table 5).

### 3.5. Associations among Lifestyle Variables and Media Utilization and Strong Health-Related Radiation Anxiety Groups

In a multivariate logistic regression analysis, Model 1 compared the continuing/emerging strong radiation anxiety and no strong radiation anxiety group. In addition, we conducted a multivariate logistic regression analysis adding the media utilization to obtain information about radiation (local, national media, public broadcasting, internet media, and public relations information from local governments) and lifestyle variables (regular exercise habits, satisfaction with sleep quality, and appropriate drinking) as independent variables. These variables were adjusted for age, gender, educational background, and area in Fukushima Prefecture.

Model 2 compared between the improved strong radiation anxiety and no strong radiation anxiety groups while adding the independent variables of media utilization and lifestyle (each analysis was adjusted for age, gender, educational background, and area in Fukushima Prefecture).

As a result, only the local media utilization variable showed a significant association with lower continuing/emerging strong radiation anxiety in the Model 1 analysis (OR: 0.435, 95% confidence interval (CI): 0.21–0.90). Furthermore, there was no significant association between the lifestyle variables related to health-promoting activities and strong radiation anxiety in Models 1 and 2 (Table 6).

### 3.6. Relationship between Psychological Distress and Post-Traumatic Stress among Those Who Have Continuing/Emerging Strong Radiation Anxiety

We conducted an analysis of covariance (ANCOVA) stratified by post-traumatic stress levels to examine the relationship between psychological distress and post-traumatic stress among those who have continuing/emerging strong radiation anxiety. We excluded two respondents who did not answer the post-traumatic stress (four-item PCL) questionnaire. This analysis was adjusted for age, gender, educational background, and living area in Fukushima. For stratifying responses by post-traumatic stress levels, the scores of the four-item PCL were divided into two groups: eight points and more, and less than eight points. A score of eight points was obtained if participants answered “a little” to all four questions. Incidentally, the third quartile of the four-item was seven, and the mean was 5.81.

As a result of the ANCOVA, the K6 score, representing psychological distress, among respondents with continuing/emerging strong radiation anxiety was 8.13 (SD: 1.06). In the responses stratified by post-traumatic stress, the K6 score was 12.63 (SD: 1.66) among respondents with higher post-traumatic stress. Meanwhile, the score was 5.13 (SD: 1.05) among respondents with lower post-traumatic stress. There was a significant difference between them (*p* = 0.004, (F(1, 350) = 9.33)) (Figure 4).

## 4. Discussion

The present study has revealed the following findings. First, regarding the association between media utilization and lifestyle variables related to health-promoting activities and continuing/emerging or improved strong radiation anxiety, there was only a significant association between local media utilization and continuing/emerging strong health-related radiation anxiety. Second, among the respondents with continuing/emerging strong radiation anxiety, psychological distress among respondents with high post-traumatic stress was significantly higher than those with low post-traumatic stress.

### 4.1. Association between Media Utilization and Lifestyle Variables Related to Health-Promoting Activities

In our previous study, there was a significant association between lowered health-related radiation anxiety and the use of local TV broadcasts and public relations information from the local government 5 years after the accident [11,12]. Regarding local TV broadcasts, it may be that by conveying information based on detailed local data in a way that addressed residents’ concerns, private local television stations could have reduced the anxiety levels among those who used them as a source of information [11]. In the present study, there was only a significant association between any local media utilization and continuing/emerging strong health-related radiation anxiety. The local newspapers (e.g., *Fukushima Minpo* and *Fukushima Minyu*) and also local TV broadcasts have continued to provide a response after the accident and during reconstruction of the evacuation or ex-evacuation areas [27,28]. Therefore, they may have had an influence, reducing the risk of continuing/emerging strong health-related radiation anxiety among residents of the Fukushima Prefecture.

Conversely, regarding public relations information from local governments, the number of articles relating to the Fukushima Daiichi nuclear power plant accident has gradually decreased after the evacuation orders were lifted [29]. The public relations information from the local government in the evacuation areas included several articles regarding records of decontamination processes, records of discussions on promoting health risk communications, and articles providing information such as general health consultations and dialogue with evacuees and experts, as well as information including education regarding stress reactions and coping [12]. Since public relations information from local governments could work positively on reducing radiation anxiety, the significant association between strong radiation anxiety and utilization of public relations information from local governments may disappear. Thus, the decreasing number of articles related to the accident in public relations information may cause the association with strong radiation anxiety to disappear.

According to previous studies conducted following the Fukushima Daiichi nuclear power plant accident, participating regularly in physical activity was significantly associated with maintaining a healthy mental health status among employees, even in harsh workplace conditions [17]. Moreover, regular physical activity is associated with the promotion of mental health recovery [18]. According to the multivariate analysis, subjective sleep insufficiency and drinking problems 3 years after the accident were associated with the severity of psychological distress [30]. Those who began drinking after the accident had serious psychological distress that may have caused depression or anxiety disorders [31,32]. Therefore, we hypothesized that undesirable lifestyle factors, such as poor-quality sleep, no regular exercise, and inappropriate drinking, may cause continuing/emerging strong health-related radiation anxiety. However, the results did not show a significant association with continuing/emerging strong radiation anxiety, although physical activity, sleep quality, and drinking behavior had a significant association with psychological distress in previous studies. Our findings may indicate that health-promoting activities related to the disaster may not address long-term and lasting strong health-related radiation anxiety, and other approaches may be needed to address continuing/emerging strong radiation anxiety.

### 4.2. Relationship between Psychological Distress and Post-Traumatic Stress among Those Who Have Continuing/Emerging Strong Radiation Anxiety

For respondents with continuing/emerging strong radiation anxiety, psychological distress among the respondents with high post-traumatic stress was significantly higher than those with low post-traumatic stress. Although this study was a cross-sectional study and it was impossible to discover the causal relationship, our findings imply that traumatic events, such as the nuclear power plant accident, and subsequent post-traumatic stress could strengthen the residents’ psychological distress.

Previous studies following nuclear power plant accidents have reported increasing rates of poor self-rated health among the general population as well as clinical and subclinical levels of post-traumatic stress disorder. These long-term mental health consequences continue to be a concern 25 years after the Chernobyl nuclear power plant disaster [33]. After the Fukushima Daiichi nuclear power plant accident of 2011, the proportion of residents with severe traumatic symptoms remained high even 2 years after the accident (21.6% in February 2012 and 18.3% in February 2013) [34]. The continued fears and uncertainties about health due to the effects on mental health of possible radiation could have raised the percentage of persons already showing strong PTSD symptoms. [35]. Furthermore, our previous study reported that the levels of psychological distress, post-traumatic stress, and radiation health anxiety were found to still be high among residents who had rebuilt permanent homes after the evacuation, with significant differences compared with those who did not evacuate [3]. Moreover, higher risk perceptions of radiation exposure after the nuclear accident were associated with later post-traumatic stress symptoms [36]. Therefore, it was thought that in traumatic experiences such as a nuclear power plant accidents, subsequent remaining post-traumatic stress in the long-term may cause continuing radiation anxiety.

The findings in the present study showed that residents with continuing/emerging strong radiation anxiety showed high psychological distress, which has been strengthened by higher post-traumatic stress. Thus, continuing strong radiation anxiety may be caused by higher post-traumatic stress; consequently, it may cause a higher psychological distress status.

Considering our findings (e.g., no association between lifestyle variables and strong radiation anxiety), it may be necessary for those who have continued strong radiation anxiety to be provided specific support from a perspective of trauma care rather than a health promotion approach. Therefore, other approaches are necessary to connect people to specialized mental health care facilities (e.g., using gatekeepers for radiation health anxiety and referrals to mental health and welfare centers) [37].

### 4.3. The Limitations and Strengths of the Present Study

This study has several limitations. First, due to its cross-sectional design, causality could not be established. Specifically, the four-item PCL was used as an outcome of the experience of the traumatic event. However, it was impossible to indicate the causality since both psychological distress (K6 score) and post-traumatic stress (four-item PCL) represented the current mental health status. Although it was uncertain how the experience of the traumatic event and any subsequent post-traumatic stress was involved in continuing/emerging strong health-related radiation anxiety, our findings may show that there was a significant relationship between high psychological distress and higher post-traumatic stress among residents who have continued or emerging strong radiation anxiety. Second, the number of respondents with emerging strong health-related radiation anxiety was extremely small (*n* = 6). Therefore, it was impossible to conduct an analysis for this group alone. Third, our primary independent variable (i.e., post-traumatic stress, measured by the four-item PCL) has not yet been validated in a Japanese context. Therefore, further studies are needed to confirm the validity and reliability of the Japanese version of the four-item PCL. Fourth, those who have continued or emerging strong health-related radiation anxiety included non-evacuees (e.g., those from Hama-Dori, Naka-Dori, and Aizu area) because of the small number of evacuees (*n* = 17). However, even if they lived in a non-evacuation area during the accident, some residents have continued or emerging strong radiation anxiety. Therefore, we analyzed all four areas in this study. The fifth limitation was that recall effects should be considered since we used a self-administered questionnaire survey. The subjects answered for themselves regarding their current situation concerning most queries at the point of the survey, except “radiation health risk perception at the time of the accident”. However, this question provided crucial information for this study. Therefore, some care should be taken in interpreting the findings. Finally, the response rate was less than 50%, presenting an issue regarding the representativeness of the analyzed group. Previous studies have reported that mental health status might affect the response rate to a survey, suggesting that a non-response might be related to one’s mental health status [38]. Many evacuees with poor mental health status may not have been able to answer the survey. This could have led to underestimations in our analysis. However, our survey may provide beneficial data because the response rate of other similar surveys conducted in evacuation areas was much lower.

Despite these limitations, our findings indicate that those who have continued or emerging radiation anxiety showed high psychological distress, which has been strengthened by the experience of the traumatic event and subsequent post-traumatic stress, based on a relatively large-scale community-based questionnaire survey. Furthermore, it may be necessary for those who have continued strong radiation anxiety to be provided with specific support from a perspective of trauma care rather than a health promotion approach. Our findings may provide important lessons for resolving long-lasting strong health-related radiation anxiety [39].

## 5. Conclusions

Nine years after the Fukushima Daiichi nuclear power plant accident, we conducted a community-based questionnaire survey to examine the factors related to long-lasting, strong health-related radiation anxiety. Although there was no significant association between lifestyle variables and media utilization, except for local media, our findings showed that those who have continuing or emerging strong radiation anxiety had high psychological distress, which may have been strengthened by the experience of the traumatic event and post-traumatic stress. Therefore, our findings imply that those who have continued strong radiation anxiety need to be provided with specific trauma care.

## Figures and Tables

**Figure 1 ijerph-18-12048-f001:**
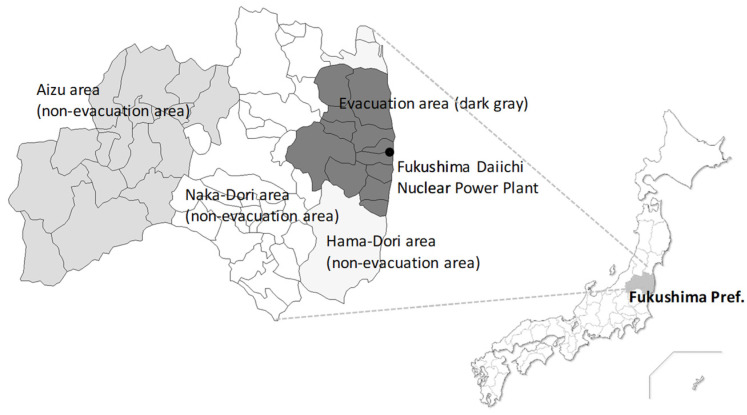
Areas within Fukushima Prefecture (evacuation area, and Hama-Dori, Naka-Dori, and Aizu).

**Figure 2 ijerph-18-12048-f002:**
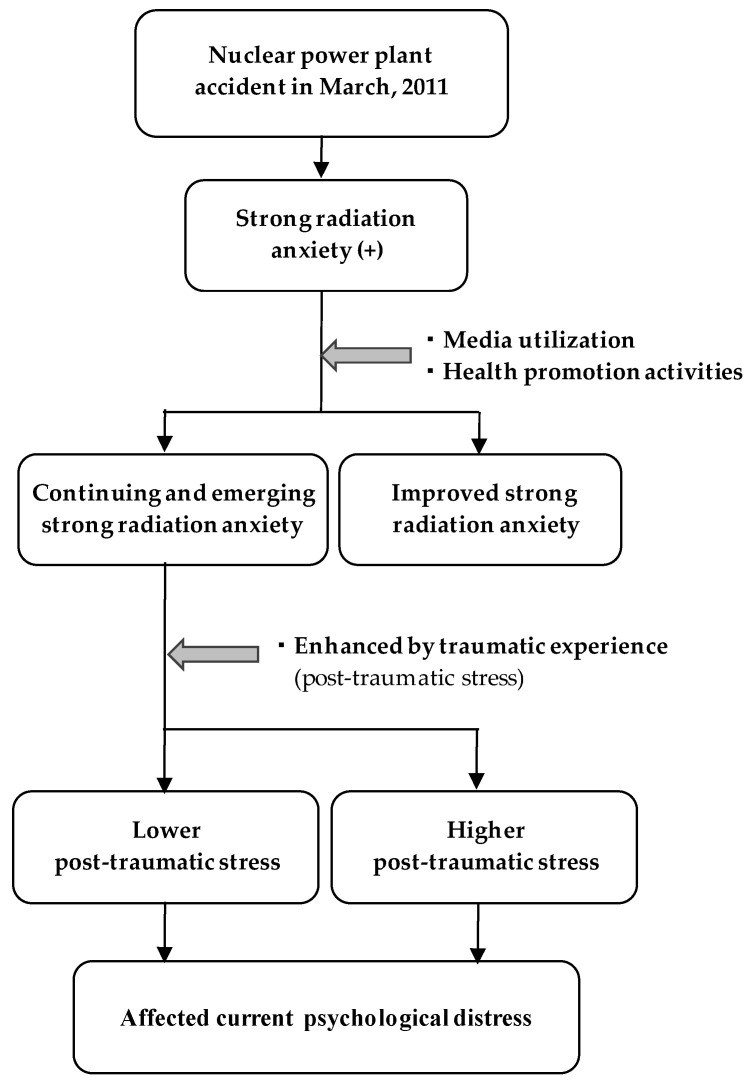
Conceptual model of the analysis in present study.

**Figure 3 ijerph-18-12048-f003:**
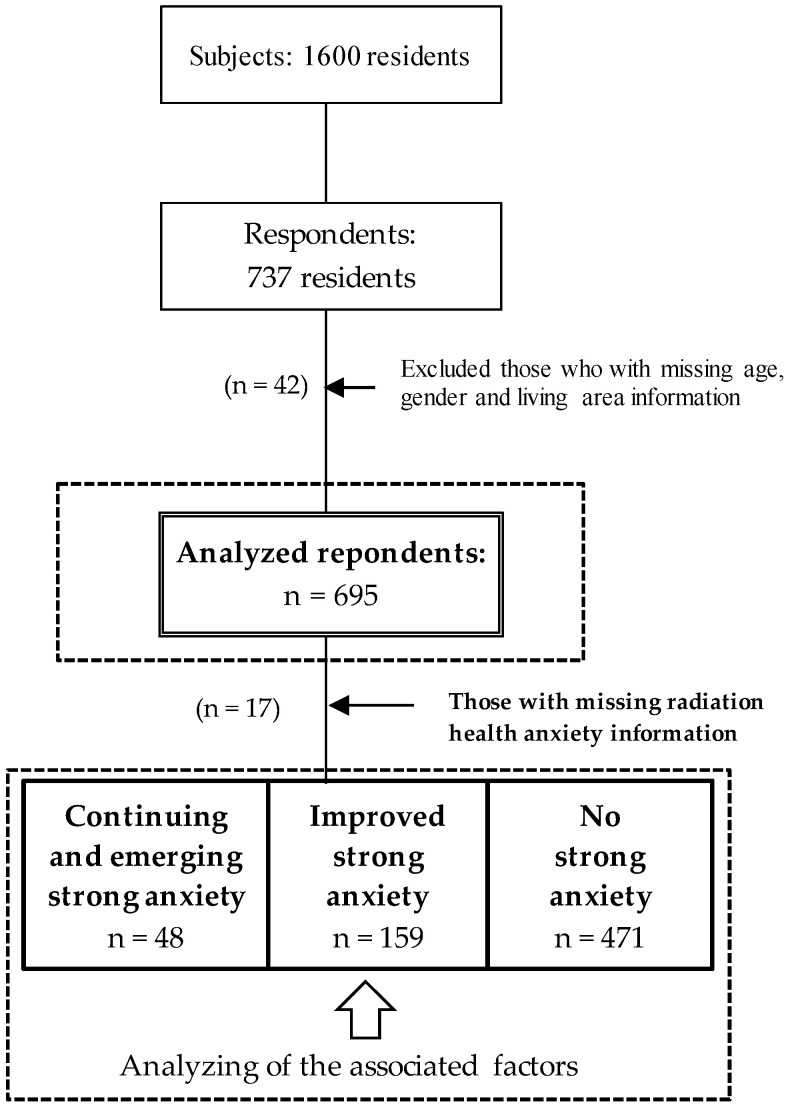
Sample selection and analysis of subjects in Fukushima Prefecture.

**Figure 4 ijerph-18-12048-f004:**
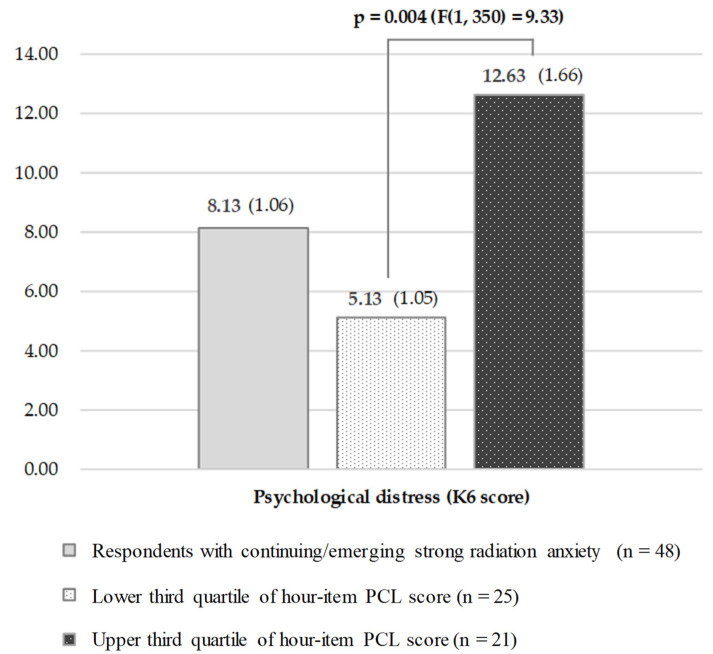
Psychological distress among those who have continuing/emerging strong radiation anxiety, stratified by post-traumatic stress levels: K6 score (stratified by post-traumatic stress), adjusted by age, gender, educational background, and living area in Fukushima. Parentheses ( ): SD, standard deviation.

**Table 1 ijerph-18-12048-t001:** Basic characteristics of respondents.

	Respondents (*n* = 695)
**Age category**	
<40 years	93 (13.4%)
40–64 years	271 (39.0%)
≥65 years	331 (47.6%)
**Gender**	
Men	336 (48.3%)
Women	359 (51.7%)
**Educational background**	
Junior or senior high school	434 (64.1%)
Vocational college, university, graduate school	249 (35.9%)
**Living area in Fukushima**	
Evacuation area	138 (19.9%)
Hama-Dori area	172 (24.7%)
Naka-Dori area	195 (28.1%)
Aizu area	190 (27.3%)

**Table 2 ijerph-18-12048-t002:** Media utilization, lifestyle variables, and current mental health status of respondents.

		Respondents (*n* = 695)
**Media utilization for information about radiation**		
Local media	Local newspaper	414 (59.6%)
	Local broadcasting	279 (40.1%)
	Any local media	526 (75.7%)
National media	National newspaper	95 (13.7%)
	National broadcasting	202 (29.1%)
	Any national media	277 (39.9%)
Public broadcasting (NHK)		377 (54.2%)
Internet media	Internet news	169 (24.3%)
	Other internet sites	31 (4.5%)
	Social network service	36 (5.2%)
	Any internet media	198 (28.5%)
Public relations information from local government		207 (29.8%)
**Lifestyle variables**		
Regular exercise	More than twice a week	72 (10.4%)
Sleep quality	Satisfied	226 (32.8%)
Drinking behavior	Appropriate drinking	202 (29.4%)
**Current mental health status**		
Psychological distress	K6 score (mean, SD)	4.08 (0.18)
Post-traumatic stress	PCL (mean, SD)	5.82 (0.10)

**Table 3 ijerph-18-12048-t003:** Categorization of continuing, emerging, improved, and no strong health-related radiation anxiety.

Respondents (*n* = 672)		Health-Related Radiation Anxiety at the Time of the Accident				
		Extremely	Very	Somewhat	Only a Little	Not at All
Current health-related radiation anxiety	Extremely	** * 18 * **	** * 1 * **	** * 1 * **	** * 1 * **	** * 1 * **
	Very	** * 7 * **	** * 16 * **	** * 3 * **	** * 0 * **	** * 0 * **
	Somewhat	**40**	**53**	99	7	6
	Only a little	**11**	**33**	75	59	13
	Not at all	**14**	**8**	54	59	99

Italics and underlining indicate continuing and emerging strong health-related radiation anxiety. Bold text indicates improved strong health-related radiation anxiety.

**Table 4 ijerph-18-12048-t004:** Basic characteristics of continuing, emerging, improved, and no strong radiation anxiety for health.

	Continuing/Emerging Strong Radiation Anxiety (*n* = 48)	Improved Strong Radiation Anxiety (*n* = 159)	No Strong Radiation Anxiety (*n* = 471)	*p*-Value
**Age category**				
<40 years	11 (22.9%)	18 (11.3%)	64 (13.6%)	
40–64 years	17 (35.4%)	73 (45.9%)	175 (45.2%)	
≥65 years	20 (41.6%)	68 (42.8%)	232 (49.3%)	*p* = 0.110 (χ^2^ = 7.54)
**Gender**				
Men	23 (47.9%)	51 (32.1%)	251 (53.3%)	
Women	25 (52.1%)	108 (67.9%)	220 (46.7%)	*p* < 0.001 (χ^2^ = 8.25)
**Educational background**				
Junior or senior high school	33 (68.8%)	86 (54.1%)	311 (66.3%)	
Vocational college, university, graduate school	15 (31.3%)	73 (45.9%)	158 (33.7%)	*p* = 0.016 (χ^2^ = 8.25)
**Living area in Fukushima**				
Evacuation area	17 (38.1%)	35 (22.0%)	82 (17.4%)	
Hama-Dori area	12 (23.8%)	46 (28.9%)	108 (22.9%)	
Naka-Dori area	16 (35.7%)	49 (30.8%)	127 (27.0%)	
Aizu area	3 (4.8%)	29 (18.2%)	154 (32.7%)	*p* < 0.001 (χ^2^ = 28.0)

**Table 5 ijerph-18-12048-t005:** Simple tabulation of media utilization, lifestyle variables, and current mental health status of those with continuing/emerging, improved, and no strong health-related radiation anxiety.

		Continuing/Emerging Strong Radiation Anxiety (*n* = 48)	Improved Strong Radiation Anxiety (*n* = 159)	No Strong Radiation Anxiety (*n* = 471)	*p*-Value
**Media utilization for information about radiation**					
Local media	Local newspaper	26 (54.2%)	92 (57.9%)	286 (60.7%)	*p* = 0.596 (χ^2^ =1.03)
	Local broadcasting	15 (31.3%)	61 (38.4%)	199 (42.3%)	*p* = 0.272 (χ^2^ = 2.60)
	Any local media	31 (64.6%)	120 (75.5%)	363 (77.1%)	*p* = 0.156 (χ^2^ = 3.72)
National media	National newspaper	5 (10.4%)	26 (16.4%)	62 (13.2%)	*p* = 0.473 (χ^2^ = 1.50)
	National broadcasting	17 (35.4%)	41 (25.8%)	142 (30.1%)	*p* = 0.376 (χ^2^ = 1.96)
	Any national media	20 (41.7%)	61 (38.4%)	192 (40.8%)	*p* = 0.849 (χ^2^ = 0.33)
Public broadcasting (NHK)		20 (41.5%)	87 (54.7%)	262 (55.6%)	*p* = 0.180 (χ^2^ = 3.43)
Internet media	Internet news	13 (27.1%)	41 (25.8%)	114 (24.2%)	*p* = 0.858 (χ^2^ = 0.31)
	Other Internet sites	1 (2.1%)	11 (6.9%)	19 (4.0%)	*p* = 0.223 (χ^2^ = 3.00)
	Social network service	2 (4.2%)	8 (5.0%)	25 (5.3%)	*p* = 0.940 (χ^2^ = 0.12)
	Any Internet media	14 (29.2%)	53 (33.3%)	129 (27.4%)	*p* = 0.360 (χ^2^ = 2.05)
Public relations information from local governments		13 (27.1%)	54 (34.0%)	136 (28.9%)	*p* = 0.450 (χ^2^ = 1.60)
**Lifestyle variables**					
Regular exercise	More than twice a week	2 (4.3%)	17 (10.9%)	50 (10.7%)	*p* = 0.370 (χ^2^ = 2.00)
Sleep quality	Satisfied	16 (34.8%)	47 (29.7%)	153 (32.7%)	*p* = 0.731 (χ^2^ = 0.63)
Drinking behavior	Appropriate drinking	34 (72.3.%)	121 (77.1%)	319 (68.3%)	*p* = 0.110 (χ^2^ = 4.42)
**Current mental health status**					
Psychological distress	K6 score (mean, SD)	8.13 (1.06)	4.61 (0.34)	3.45 (0.19)	*p* < 0.001 (F = 25.3)
Post-traumatic stress	Four-itemPCL (mean, SD)	7.70 (0.60)	6.27 (0.22)	5.46 (0.10)	*p* < 0.001 (F = 20.5)

**Table 6 ijerph-18-12048-t006:** Multivariate logistic regression analysis of media utilization and lifestyle variables related to health-promoting activities with current continuing/emerging, improved, and no strong health-related radiation anxiety.

	Continuing/Emerging Radiation Anxiety (Ref.: No Radiation Anxiety)			Improved Radiation Anxiety (Ref.: No Radiation Anxiety)		
	Odds Ratio *	95% CI	*p*-Value	Odds Ratio *	95% CI	*p*-Value
**Media utilization for information about radiation**						
Any local media	0.435	(0.21–0.90)	0.025	1.020	(0.63–1.65)	0.937
Any national media	0.789	(0.39–1.60)	0.510	0.964	(0.63–1.47)	0.866
Public broadcasting (NHK)	0.623	(0.32–1.22)	0.170	1.064	(0.70–1.60)	0.769
Any internet media	0.585	(0.25–1.36)	0.213	1.360	(0.82–2.24)	0.229
Public relations information from local governments	0.789	(0.36–1.71)	0.549	1.310	(0.83–2.06)	0.242
**Lifestyle variables**						
Regular exercise habits	0.341	(0.78–1.50)	0.154	1.095	(0.59–2.03)	0.775
Sleep satisfaction	1.381	(0.70–2.71)	0.347	0.901	(0.60–1.36)	0.621
Appropriate drinking	1.126	(0.54–2.37)	0.754	0.912	(0.57–1.46)	0.701

Odds ratio *: adjusted by age, gender, educational background, and living area in Fukushima. 95% CI: 95% confidence interval.

## Data Availability

Data underlying the findings in this study cannot be made publicly available due the nature of ethical approval for the study. Interested researchers may submit requests to the Fukushima Medical University’s Ethics Committee (Contact information: Email: rs@fmu.ac.jp) for access to confidential data.

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
