# Peer review of "Those Who Have Continuing Radiation Anxiety Show High Psychological Distress in Cases of High Post-Traumatic Stress: The Fukushima Nuclear Disaster"

_ijerph, 2021, doi:10.3390/ijerph182212048_

Round 1

Reviewer 1 Report

In the introduction, it is unclear the section that talks about information disaster and media utilization (line 55 -62). Can the authors ensure more clarity?

Figure 2 last box, "Affected on current psychological distress" maybe replace with "Affected current psychological distress". Also should this be "Improved strong radiation anxiety" replaced with "Increased strong radiation anxiety"? Similarly, line 241 should be replaced with "increased" and not improved.

Authors to check line 323. There is an empty bracket.

Line 425-427 "Furthermore, it may be necessary for those who have continued strong radiation anxiety to provide specific support with a perspective of trauma care rather than a health promotion approach". Are the authors referring to "receive" instead of "provide"?

Line 195-196 - Can authors provide further explanation to why the 42 were excluded?

Line 50- replace  statuses with "status".

Author Response

Thank you for your courteous comments. Could you please check the attachted file?

Reviewer 2 Report

Dear authors,

Your study - "Those who have been continuing radiation anxiety show high psychological distress and strengthened post-traumatic stress: A Fukushima nuclear disaster" is very interesting and excellent research. Maybe you can think about rephrasing your title to be more precise.

Without denying the significance or quality of your study, I would like to bring out a few flaws that should be addressed before the article is published. First, I propose that the paper's introduction section be split into an introduction and a literary review. Add additional research articles from a regional viewpoint to the literary review. I would recommend improving the discussion with more comparing previous research. The conclusion is inadequate, and all of its parts must be improved. Also, I would recommend including a survey questionnaire in the paper.

Respectfully

Author Response

(The authors gave the same response as above.)

Round 2

Reviewer 2 Report

Dear authors,

Thank you for your improvements.

Kind regards